# Characterisation of Tannin and Aroma Profiles of Pinot Noir Wines Made with or without Grape Pomace

Pradeep M. Wimalasiri [1], Jicheng Zhan [2] and Bin Tian [1,*]

1    Faculty of Agriculture and Life Sciences, Lincoln University, Lincoln 7647, New Zealand
2    Beijing Key Laboratory of Viticulture and Enology, College of Food Science and Nutritional Engineering, China Agricultural University, Beijing 100083, China
*    Correspondence: bin.tian@lincoln.ac.nz; Tel.: +64-3-423-0645

**Abstract:** The contribution of grape pomace on tannin concentration, tannin composition and aroma profile of Pinot noir wine was studied using different fermentation media to make up four treatments: GJ-P, grape juice plus pomace; MJ-P, model juice plus pomace; GJ, grape juice; MJ, model juice. The MJ-P treatment showed significantly lower amounts of tannins, mean degree of polymerisation (mDP), similar amounts of anthocyanin, and a similar secondary aroma profile compared to the GJ-P treatment. Grape pomace addition significantly increased the tannin concentration in wines. This study was also revealed the importance of phenolics present in grape juice in tannin polymerisation and final tannin concentration in wines. Grape pomace addition significantly reduced some important aroma compounds such as acetate esters (except ethyl acetate), most of the volatile fatty acids, a few ethyl esters and β-damascenone but increased some primary aromas in wines due to the presence of their aroma precursors in skins. Hence, these results indicate that grape pomace may bind or delay the release of some aroma compounds and/or lose these compounds during cap management in GJ-P and MJ-P treatments compared to the respective juice treatments.

**Keywords:** aroma profile; grape pomace; model juice; Pinot noir; tannin

## 1. Introduction

Tannins, anthocyanin, and the aroma profile of red wines collectively determine the final wine quality. Wine phenolics mainly originated from grape skins, seeds, and stems if used in the winemaking process, of which extraction could be influenced by various winemaking techniques [1–3]. Volatile aroma compounds in wine are usually divided into three groups based on their origin: primary aroma compounds are derived from grapes; secondary aroma compounds are mainly originated from the metabolism of yeast and bacteria; and tertiary aroma compounds are formed from barrel maturation or bottle ageing [4,5].

In winemaking, tannin origin, composition, and extractability determine the final phenolic and colour attributes of the wine [6–8]. Ultimately, wine colour, flavour and mouthfeel are largely determined by the relative proportions of tannin extracted from different grape tissues in the winemaking process. Making wine from Pinot noir grapes has long been a challenge due to its delicate flavour, light colour, and sometimes poor ageing potential of the wine [2]. According to previous studies, Pinot noir grapes have a higher seed tannin concentration compared to skin tannin levels, which results in a lower skin-to-seed tannin ratio [2,9,10]. This tannin distribution in Pinot noir berries may contribute to lower tannin concentration in wine and increase seed-originated tannin proportion in the wine [11,12]. Previous studies suggested that the perception of astringency in wine is related to berry tannin concentration, composition, and molecular size of the tannins. Moreover, a greater degree of polymerization and a greater percentage of galloylation (G%), are reported to provoke the sensation of astringency in wine, and lower molecular weight

tannins could contribute to bitterness in wines [6,13,14]. Hence, the extraction of optimum proportions of tannins from different tissues is important in wine quality. In the present study, wines prepared with grape juice and model juice with and without grape pomace help to understand the effect of grape juice and pomace on final wine tannin concentration and composition.

The factors affecting the formation of wine aromas are complex. In general, wine aroma is greatly influenced by the grape variety, relative proportions of grape skins, vintage, viticultural practices, maceration and winemaking practices, regional variation, etc. [15–19]. During fermentation, yeast converts grape sugars into alcohol and fermentation drives complex chemical reactions that affect the aroma and flavour of the wine. Primary aromas such as monoterpenes, $C_{13}$-norisoprenoids, methoxypyrazines, some volatile phenols such as eugenol, $C_6$ compounds and thiols are mainly found in grape skin, in free volatile form or bound into non-volatile precursors such as glycoside. Later, glycosidically bounded aroma compounds are released during fermentation or ageing and contribute to primary aroma and wine quality [15,20]. Most of the previous studies have also confirmed that there is a positive correlation between skin contact time and the concentration of primary aromas observed in final wine [21]. In contrast, fermentation-derived aroma compounds make up the largest percentage of the total aroma composition of wine comprising volatile fatty acids, higher alcohols, ethyl esters and acetate esters but most of them do not significantly contribute to the wine aroma due to their higher aroma thresholds in wine [21]. However, the dividing line between primary aromas and secondary aromas is not clear-cut in terms of their origin. For example, yeast also could indirectly contribute to the primary aromas by converting odourless grape-derived precursors into important aroma compounds through enzymatic hydrolysis in wine [22].

There is very limited studies investigating the effect of grape pomace and juice on Pinot noir wine composition. In this study, Pinot noir grape pomace and juice were separated, and model juice was used to substitute the grape juice, in order to have a better understanding on the contribution of individual grape components (grape juice and grape pomace) to the tannin and aroma composition of resultant Pinot noir wine.

## 2. Materials and Methods

### 2.1. Chemicals

All chemicals used for model juice preparation, phenolic analysis and HPLC were purchased from Sigma-Aldrich (Auckland, New Zealand). Purchased aroma standards used in GC-MS include ethyl decanoate and *cis*-3-hexen-1-ol (Fluka Chemicals, Ronkonkoma, NY, USA), deuterated benzaldehyde-d6 and hexanoic-d11-acid (Isotec, St. Louis, MO, USA). The deuterated octanoic-d2-acid, isoamyl-d3-acetate, ethyl-d5-hexanoate, hexyl-d3-acetate, ethyl-d5-octanoate, d5-ethyl butyrate, and d5-ethyl decanoate were previously synthesized and stored at Lincoln University, New Zealand [17].

### 2.2. Microvinification

Pinot noir grapes were harvested in 2020 from the Akarua vineyard in Central Otago, New Zealand. In this study, four treatments of wines were prepared through fermentation of: (1) grape juice and pomace (GJ-P), (2) model juice and grape pomace (MJ-P), (3) grape juice (GJ), and (4) model juice (MJ). Winemaking in each treatment was triplicated in 2 L plastic containers.

Hand-destemmed grapes (1.6 kg) were used in each replicate of the GJ-P treatment. Another 1.6 kg of hand-destemmed grapes was pressed to obtain 1 L of grape juice that was used for each replicate of the GJ treatment, and the pressed grape pomace was combined with 1 L of the model juice to prepare each replicate of the MJ-P treatment. Each replicate of the MJ treatment was prepared using 1 L of model juice without pomace.

Model juice included three different solutions, which were sterilized separately at 121 °C for 20 min and then combined aseptically [23]: solution A (110 g/L D-glucose, 110 g/L D-fructose, 10 mg/L ergosterol, and 1 mL/L Tween 80), solution B (6 g/L L-(+)-

tartaric acid, 3 g/L L-(−)-malic acid, and 0.5 g/L citric acid), and solution C (1.7 g/L yeast nitrogen base with amino acids, 2 g/L casamino acids, 0.2 g/L $CaCl_2$, 0.8 g/L arginine-HCl, 1 g/L L-(2)-proline, and 0.1 g/L L-(2)-tryptophan). The final pH was adjusted to 3.21 using 2 M sodium hydroxide to match the grape juice.

The EC1118 yeast (Lallemand, Montreal, QC, Canada) was inoculated at 0.25 g/L for all winemaking treatments. The fermentation temperature was maintained in the range of 25 to 30 °C. The weights of ferments were recorded to monitor the progress of fermentation. For the fermentations with addition of pomace, the cap was plunged twice a day until the end of fermentation. Free-run wines were then collected and added with 50 mg/L of $SO_2$. After two days of cold settling at 4 °C, wine samples were racked off the lees and bottled.

### 2.3. Oenological Parameters

Total soluble solids (TSS), residual sugar, alcohol content, pH, and titratable acidity were measured using standard protocols [24]. Briefly, the total soluble solid (TSS) in the juice was analysed using an Atago 101 digital handheld refractometer with an automatic temperature compensation function (Atago Co., Ltd., Tokyo, Japan). The Rebelein method was used to determine the residual sugar concentration in wines. Alcohol concentration was measured by an ebulliometer (Dujardin Salleron, France). Titratable acidity and pH were determined using a Suntex SP-701 pH meter. Titratable acidity (TA) was measured by titrating 10 mL of juice/wine with a standardised 0.1 N sodium hydroxide (NaOH) solution to a pH endpoint of 8.2. Yeast assimilable nitrogen (YAN) was determined using a commercial enzyme test kit (Vintessential Laboratories, Dromana, Victoria, Australia).

### 2.4. Spectrophotometric Analysis

Wine samples were centrifuged at $3000 \times g$ for 5 min (Model Heraeus Multifuge X1R, Thermo Fisher Scientific, Lower Saxony, Germany) prior to analysis. Spectrophotometric measurements were carried out on a Shimadzu 1800 UV-Visible spectrophotometer (Shimadzu, Tokyo, Japan).

#### 2.4.1. Methyl Cellulose Precipitable (MCP) Tannin

Tannin in wine samples was determined using the 1 mL methyl cellulose precipitation (MCP) method [25]. The absorbance of the sample and the respective control at 280 nm was measured using the spectrophotometer. The absorbance difference between treatment and control was used to calculate the concentration of MCP tannins against an epicatechin calibration curve.

#### 2.4.2. Total Phenolics

Total phenolic content was performed using the microscale Folin–Ciocalteu assay [26]. The sample and standard solutions were measured at 765 nm using a UV–Vis spectrophotometer.

### 2.5. Analysis of Flavan-3-Ols Monomers by HPLC

Monomeric phenolic fractions in wine samples were separated using the optimised solid-phase extraction (SPE) method [27]. In brief, an Oasis HLB cartridge (3 mL, 60 mg, 30 μm) (Waters, Rydalmere, NSW, Australia) was conditioned with 2 mL of methanol followed by 2 mL of water. Wine samples were centrifuged at $3000 \times g$ for 5 min before loading into the SPE cartridge. The wine sample (1 mL) was then loaded under gravity and completely dried with a gentle stream of nitrogen. The monomeric phenolic fraction was eluted by adding 40 mL of 95% acetonitrile and 5% 0.01 M hydrochloric acid. The elute was vacuum evaporated completely at 36 °C and then dissolved in 1 mL of 10% ethanol/0.1% formic acid.

HPLC analysis was carried out according to the previously published method [28]. In brief, the monomeric phenolic fraction (10 μL) was loaded at 1 mL/min and the elution of monomeric phenolics was monitored by absorbance at 280, 320, 360 and 520 nm in the diode array detector (DAD) and the corresponding excitation at 280 nm and emission

at 320 nm in the fluorescence detector (FLD). Phenolic compounds were identified by comparing their retention times and the spectra with those of standards. Quantification of phenolic compounds was carried out by area measurements at 280 nm, 320 nm and FLD separately. Quantitative assays were achieved using external calibration curves for all standard phenolics by the dissolution of the standard solution accordingly.

### 2.6. Phloroglucinolysis

Wine tannin was characterised by acid catalysis in the presence of excess phloroglucinol followed by reversed-phase HPLC [29]. Phloroglucinolysis was carried out with modifications, which provided information on subunit composition, conversion yield, and mean degree of polymerization (mDP). Wine samples (4 mL) were rotary evaporated to 0.2–0.3 mL at 40 °C to remove ethanol, and re-dissolved in 4 mL of deionised water. After that, the sample was loaded onto a pre-conditioned 0.36 g C18 SEP-PAK cartridge (WAT051910, Global Science, Auckland, New Zealand) with 5 mL of methanol, 7.5 mL ethyl acetate and 7.5 mL of deionised water. The loaded C18 cartridge was washed with 7.5 mL of deionised water and allowed to dry with nitrogen gas at a 3 L/min flow rate for 60 min. After that, 5 mL of ethyl acetate was passed through the cartridge to wash off the monomeric phenols and the required polymeric fraction was eluted with 5 mL of methanol. This methanolic solution was rotary evaporated at 40 °C and reconstituted with methanol to a final volume of 1 mL. Then, 0.5 mL of this prepared methanolic solution was reacted with an equal volume of phloroglucinol reagent (A solution of 50 g/L phloroglucinol in methanol, containing 0.1 M hydrochloric acid, and 10 g/L ascorbic acid) in a screw cap tube in a water bath at 50 °C for 20 min. To stop the reaction, 5 mL of 40 mM sodium acetate solution was added into the screw cap tube and vortexed the mixture for dissolving. This solution was filtered through a 0.45 μm pore size 13 mm diameter PTFE filter into the HPLC vial (the first few drops discarded).

Reversed-phase HPLC analysis was carried out using an Agilent 1100 series HPLC (Waldbronn, Germany) equipped with a quaternary pump, diode array detector (DAD) and fluorescence detector (FLD) was used to identify and quantify proanthocyanidin cleavage products. A sample fraction of 40 μL was injected and separated using an ACE C18-PFP 150 × 4.6 mm, particle size 3 μm column (Advanced Chromatography Technologies, Aberdeen, Scotland) that was thermostat at 25 °C. Two mobile phases were used: 2% *v/v* aqueous acetic acid (A), and 2% *v/v* acetic acid in methanol (B). The flow rate was set at 0.8 mL/min, with the mobile phase set initially as a linear pump gradient from 5 to 10% B in 5 min, a gradient from 10 to 40% B in 25 min, and a gradient from 40 to 100% B in 1 min and held for 5 min, then reduced to 5% B in 1 min and held for 5 min, before equilibrating the column to the initial running conditions. The detection and quantification of proanthocyanidin cleavage products were conducted at 280 in the DAD. The corresponding excitation at 280 nm and emission at 320 nm in the FLD were used for assisting in the identification and confirmation.

To estimate the molarity of proanthocyanidin cleavage products (terminal and extension subunits), the areas obtained from HPLC analysis were divided by their respective molar response factors (relative to (+)-catechin) [29] to convert into catechin equivalent molar response areas. Then, the catechin equivalent molar response areas were used to calculate the molarity of proanthocyanidin cleavage products using a catechin calibration curve. After that, the relative molar composition of terminal subunits (flavan-3-ol monomers) and extension subunits (phloroglucinol adducts) were calculated separately. The mDP was calculated as the sum of all subunits (terminal and extension subunits in moles) divided by the sum of all terminal subunits (in moles). The per cent conversion yield was determined as the proportion of epicatechin equivalent total tannin concentration determined by phloroglucinolysis divided by the epicatechin equivalent total tannin results from the MCP tannin assay.

### 2.7. Volatile Aroma Compounds

Volatile aroma compounds were analysed according to the methods described previously [17,30]. Three different methods were used to determine three groups of aroma compounds: esters and alcohols, volatile fatty acids, and low-concentration compounds.

The wine sample (0.9 mL) was pipetted together with 8.06 mL of pH 3.5 acidified water, followed by 40 μL of deuterated internal standard solution and 4.5 g of sodium chloride into a 20 mL SPME vial. The samples were incubated and agitated for 10 min at 60 °C. The SPME fibre (Stableflex DVB/CAR/PDMS, Sigma-Aldrich, St. Louis, MO, USA) was conditioned for 60 min at 60 °C. Desorption of the fibre occurred in the injection port at 270 °C for 5 min. The GC–MS was equipped with dual columns in series: a Rtx-Wax column (30.0 m × 0.25 mm ID × 0.5 μm film thickness, polyethene glycol, Restek, Bellefonte, PA, USA) and a Rxi–1 MS column (15 m × 0.25 mm ID × 0.5 μm film thickness, 100% dimethyl polysiloxane, Restek). Helium was used as the carrier gas with a linear velocity of 33.5 cm/s. Splitless injection was used for the initial 3 min and then switched to a 20:1 split ratio. The GC oven temperature was held at 35 °C for 3 min, heated to 250 °C at 4 °C/min and then held for 10 min. The interface and MS source were set to 250 °C and 200 °C, respectively. The MS source was operated in electron impact (EI) mode with ionization energy of 70 eV. Analysis of the chromatograms was performed on GCMS Solution software, version 2.5 (Shimadzu, Auckland, New Zealand).

For volatile fatty acids analysis, the HS-SPME extraction and GC operational conditions were changed. The helium gas flow was set at a constant linear velocity of 46.8 cm/s. The GC oven temperature was held at 50 °C for 3 min, then increased to 240 °C at 10 °C/min, further increased to 250 °C at 30 °C/min and held for 5 min. The column configuration and other operating conditions remained the same as for esters and alcohol analysis.

For low-concentration compounds analysis, the method is similar to that for esters and alcohols, but the acquisition mode was changed to the selected ion monitoring (SIM) to increase the sensitivity, and the GC oven temperature ramp was modified to 35 °C for 3 min, increased to 105 °C at 3 °C/min and held for 10 min, increased to 140 °C at 2 °C/min and held for 10 min, and finally increased to 250 °C at 4 °C/min and held for 10 min.

### 2.8. Odour Activity Values

The odour activity value (OAV) of each aroma compound was calculated as the ratio between the concentration of an individual aroma compound and the perception threshold of the corresponding compound found in the literature. All aroma compounds with OAV above 0.1 were considered significant aroma compounds in this study. Important aroma compounds were categorised into seven aroma series (fruity, floral, spicy, chemical, fatty, woody, and green) based on their odour description [28]. Woody, green, and spicy aroma series were combined to make herbaceous aroma series in this work due to the lower OAVs detected in wines.

### 2.9. Statistical Analysis

Data were presented as mean ± SD of three replicates. Data obtained from HPLC, GC-MS, and OAVs were analysed by one-way analysis of variance (ANOVA) with a post hoc analysis using the Tukey's honestly significant difference (HSD) test at a significance level of 0.05 (Minitab Inc., State College, PA, USA, version 18.1). Tannin composition data from phloroglucinolysis was analysed using a two-sample *t*-test at 0.05 level of significance using Minitab 18. All aroma compounds analysed in this study were subjected to principal component analysis (PCA). R statistical software (R Core Team 2019, Vienna, Austria) and mixOmics package from Bioconductor (https://bioconductor.org/ accessed on 21 September 2022) was used for statistical analysis.

## 3. Results and Discussion

### 3.1. Oenological Testing

At harvest, the total soluble solids (TSS) in grapes were measured at 23.4°Brix, and TSS in model juice was determined at 22.2°Brix. YAN in grape juice and model juice was measured at 236 mg/L and 362 mg/L, respectively. The pH and TA in juice and wine, alcohol content, residual sugar, total phenolics, and tannins in wine are shown in Table 1. The GJ-P wine showed significantly higher pH, total phenolics and tannin concentration compared to all other treatments. The pH and TA of the wines ranged from 3.40 to 3.16 pH and 10.91–8.54 g/L TA. Comparatively higher pH in GJ-P is likely due to the extraction of potassium from grape pomace during fermentation. However, when comparing MJ-P and GJ treatments they did not show a significant difference in pH between them because the model juice pH had to be adjusted to match grape juice pH by adding 2 M sodium hydroxide before inoculating and it ultimately caused significantly lower TA in juice and wine. The highest TA was observed in GJ treatment as expected because of the absence of grape pomace. The alcohol content ranged from 10.9 to 13.4% and reflected the differences in juice soluble solids and final residual sugar amounts in the wines.

**Table 1.** General oenological parameters.

| | GJ-P | MJ-P | GJ | MJ |
|---|---|---|---|---|
| **Juice** | | | | |
| pH | 3.20 ± 0.02 a | 3.22 ± 0.01 b | 3.13 ± 0.06 c | 3.21 ± 0.00 b |
| TA (g/L) | 9.57 ± 0.18 b | 6.38 ± 0.12 b | 9.70 ± 0.22 a | 6.10 ± 0.00 c |
| **Wine** | | | | |
| pH | 3.40 ± 0.03 a | 3.22 ± 0.02 b | 3.16 ± 0.02 c | 3.23 ± 0.02 b |
| TA (g/L) | 9.62 ± 0.19 b | 8.34 ± 0.08 c | 10.91 ± 0.58 a | 8.54 ± 0.00 c |
| Alcohol (%) | 13.4 ± 0.2 a | 10.9 ± 0.5 b | 14.2 ± 0.2 a | 11.0 ± 0.4 b |
| Residual sugar (g/L) | 2.78 ± 0.42 b | 2.22 ± 0.08 b | 1.58 ± 0.16 c | 3.70 ± 0.13 a |
| Total phenolics (mg/L) | 2705 ± 69 a | 2152 ± 34 b | 274 ± 13 c | 127 ± 3 d |
| Tannin (mg/L) | 1225 ± 119 a | 821 ± 22 b | ND | ND |

ND-Not Detected; different lowercase letters in rows indicate a significant difference among treatments ($p < 0.05$). GJ-P, grape juice plus pomace; MJ-P, model juice plus pomace; GJ, grape juice; MJ, model juice.

Comparing to MJ-P treatment, GJ-P treatment showed significantly higher MCP tannin and total phenolic concentrations in the wine. This may be due to two reasons. First, the phenolics present in grape juice, which could be involved in the polymerisation of tannins by interacting with phenolics extracted from skins [31]. Second, the higher concertation of alcohol may have increased the extraction of tannins from grape skin and seeds [32,33]. As expected, MJ and GJ treatments showed a lot lower total phenolic concertation and zero MCP tannin concertation compared to the wines fermented with grape pomace, as vast majority of the phenolics in the wine are derived from the skin and seed tissues [2,34]. The total phenolics observed in MJ treatment were mainly because the Folin–Ciocalteu method is an antioxidant assay, which measures the reductive capacity of antioxidants in the sample, such as amino acids, sugars, etc. [35].

### 3.2. Monomeric Phenolics Composition

Concentrations of monomeric phenolics determined by HPLC were shown in Table 2. The flavan-3-ols, predominantly catechin and epicatechin represented 52–54% of total monomeric phenolics quantified by HPLC in GJ-P and MJ-P treatments. Significantly higher monomeric flavan-3-ol concentrations were observed in the GJ-P treatment and reflected the same trend as that of tannin and total phenolics concentrations observed in the wines.

**Table 2.** Monomeric phenolic composition of the wines.

| Phenolics (mg/L) | GJ-P | MJ-P | GJ | MJ |
|---|---|---|---|---|
| **Flavan-3-ols** | | | | |
| Catechin | 356 ± 16 a | 306 ± 24 b | 2.40 ± 0.37 c | ND |
| Epicatechin | 95.7 ± 6.8 a | 81.8 ± 5.3 b | ND | ND |
| Procyanidin B1 | 24.2 ± 1.3 a | 17.3 ± 1.8 b | ND | ND |
| Procyanidin B2 | 21.9 ± 1.4 a | 13.4 ± 1.4 b | ND | ND |
| **Anthocyanins** | | | | |
| Delphenidin-3-*O*-glucoside | 26.7 ± 2.1 a | 31.2 ± 2.2 a | ND | ND |
| Cyanidin-3-*O*-glucoside | 1.04 ± 0.11 a | 1.21 ± 0.18 a | ND | ND |
| Malvidin-3-*O*-glucoside | 217 ± 4 a | 196 ± 22 a | 10.7 ± 0.8 b | ND |
| Peonidin-3-*O*-glucoside | 21.8 ± 1.2 a | 21.8 ± 3.5 a | 0.86 ± 0.17 b | ND |
| **Flavonols** | | | | |
| Kaempferol | 0.72 ± 0.54 ab | 1.24 ± 0.19 a | 0.14 ± 0.07 b | ND |
| Quercetin | 5.45 ± 0.78 a | 5.58 ± 0.58 a | 0.45 ± 0.20 b | ND |
| Rutin | 0.50 ± 0.02 a | 0.78 ± 0.33 a | ND | ND |
| **Hydroxybenzoic acids** | | | | |
| Gallic acid | 13.8 ± 1.1 a | 12.3 ± 0.6 a | 0.35 ± 0.05 b | 0.18 ± 0.06 b |
| Protocatechuic acid | 1.52 ± 0.04 a | 0.97 ± 0.07 b | 0.67 ± 0.07 c | ND |
| Syringic acid | 3.84 ± 1.30 a | 2.78 ± 0.94 ab | 0.95 ± 0.12 bc | ND |
| Vanilic acid | 3.82 ± 0.07 a | 1.63 ± 0.66 b | 1.27 ± 0.11 a | 3.70 ± 0.17 b |
| **Hydroxycinnamic acids** | | | | |
| Caffeic acid | 0.00 ± 0.00 b | ND | 0.09 ± 0.01 a | ND |
| Caftaric acid | 2.60 ± 0.13 b | 3.20 ± 0.09 a | 0.85 ± 0.06 c | ND |
| Cinnamic acid | 0.10 ± 0.00 b | 0.13 ± 0.02 a | ND | ND |
| Ferulic acid | 0.05 ± 0.09 b | 0.29 ± 0.04 a | 0.03 ± 0.05 b | ND |
| p-Coumaric acid | 0.00 ± 0.00 b | 0.00 ± 0.00 b | 0.28 ± 0.01 a | ND |
| **Stilbenes** | | | | |
| Resveratrol | 0.06 ± 0.05 b | 0.15 ± 0.02 a | ND | ND |

ND-Not Detected; different lowercase letters in rows indicate a significant difference among treatments ($p < 0.05$). GJ-P, grape juice plus pomace; MJ-P, model juice plus pomace; GJ, grape juice; MJ, model juice.

The main anthocyanin form found in Pinot noir wine is malvidin-3-O-glucoside [36]. Malvidin-3-O-glucoside, delphenidin-3-O-glucoside, cyanidin-3-O-glucoside, and peonidin-3-O-glucoside were quantified within the concentration range previously reported in Pinot noir wines [28,37,38]. There was no significant difference in the concentrations of monomeric anthocyanins between GJ-P and MJ-P treatment. This confirms that the model juice could extract anthocyanin from the grape skins effectively as grape juice and stabilize it in the wine matrix. A lower concentration of malvidin-3-O-glucoside was detected in GJ treatment confirming the possible leaching of anthocyanin during pressing grapes.

Flavonols in wines are mainly originated from the grape skins [39] and cluster sunlight exposure appears to be the primary factor determining flavonol levels in grapes [40]. Hence, this could be the reason for observing no significant differences in flavonol concentrations between GJ-P and MJ-P treatments because both treatments contained a similar amount of skin. Lower concentrations of kaempferol and quercetin in GJ treatment may be due to leaching from skins because they are not present in the pulp and are mainly located in grape skins. A recent study on Vranac grapes also confirmed that quercetin and kaempferol are not present in grape pulp [41].

Hydroxybenzoic acids in the GJ-P treatment, including gallic acid, protocatechuic acid, syringic acid, and vanillic acid were quantified within the concentration range previously reported in Pinot noir wine [28,42]. According to previous studies [41,43], gallic acid is mainly located in grape seeds, while syringic acid, protocatechuic acid, and vanilic acid are present in grape skins, seeds, and pulp in variable amounts. Hence, the presence of hydroxybenzoic acids (except gallic acid) in both MJ-P and GJ treatments confirms the presence of these compounds in both skins and pulp of Pinot noir berries because these compounds could quickly release from pulp to grape juice during pressing. Interestingly, the sum of hydroxybenzoic acids in MJ-P and GJ treatments is also similar to the concentrations observed in the GJ-P wine. The main hydroxycinnamic acid found in the wines was caftaric acid and it agrees with previous studies [28,44]. Hydroxycinnamic acids are mainly found in grape pulp, and also present in skin, seeds and stems in variable amounts.

Moreover, their concentration varies a lot between different tissues and different grape varieties [43,45,46]. The lower concentrations of caftaric acid, ferulic acid, and p-coumaric acid in GJ treatment confirm the presence of these compounds in grape pulp in very low concentrations in Pinot noir berries and/or these compounds are oxidised by endogenous tyrosinase when grapes are crushed [46].

### 3.3. Tannin Composition and Mean Degree of Polymerisation (mDP)

Analysis of tannin composition was carried out in GJ-P and MJ-P treatments using phloroglucinolysis (Table 3). GJ-P treatment showed significantly higher epicatechin gallate terminal and extension subunit proportions, and significantly lower epicatechin terminal and extension subunit proportions compared to MJ-P treatment. This could be mainly associated with higher seed tannin extraction during fermentation in GJ-P treatment due to the presence of higher alcohol concentration compared to MJ-P treatment [47].

**Table 3.** Tannin mean degree of polymerisation (mDP) and composition in wines.

|  | GJ-P | MJ-P |
|---|---|---|
| **Concentrations** |  |  |
| Total Terminal (nmol/L) | 270 ± 73 a | 330 ± 81 a |
| Total Extension (nmol/L) | 1368 ± 153 a | 999 ± 113 b |
| **Terminal subunit composition (%)** |  |  |
| C | 59.1 ± 0.2 a | 59.8 ± 1.2 a |
| EC | 32.1 ± 0.3 b | 33.4 ± 0.3 a |
| ECG | 8.8 ± 0.5 a | 6.9 ± 0.9 b |
| **Extension subunit composition (%)** |  |  |
| P-C | 10.4 ± 0.6 a | 11.7 ± 0.7 a |
| P-EC | 73.0 ± 0.3 b | 73.8 ± 0.1 a |
| P-ECG | 7.3 ± 0.1 a | 5.7 ± 0.6 b |
| P-EGC | 9.3 ± 0.7 a | 8.9 ± 0.2 a |
| **Tannin characteristics** |  |  |
| mDP | 6.2 ± 0.6 a | 4.1 ± 0.4 b |
| Yield | 38.7 ± 4.5 a | 47.1 ± 7.6 a |

Different lowercase letters in rows indicate a significant difference among treatments ($p < 0.05$). GJ-P, grape juice plus pomace; MJ-P, model juice plus pomace; GJ, grape juice; MJ, model juice.

GJ-P treatment showed a significantly higher mean degree of polymerisation than MJ-P treatment, agreeing with a previous study [2]. It was mainly due to the significantly higher total extension subunit concertation recorded in GJ-P treatment because the phenolic acids in grape juice and anthocyanins leached into grape juice may contribute to the tannin polymerisation reactions in GJ-P treatment. Many previous studies have shown that anthocyanin is associated with polymerisation reactions with tannins present in the must during winemaking [31,48,49].

### 3.4. Aroma Compound Analysis

Forty-six aroma compounds were identified and quantified in the wines, including nineteen esters, seven volatile fatty acids, eight higher alcohols, one aldehyde, four volatile phenols, three C13-norisoprenoids and four monoterpenes (Table 4). All aroma compounds showed significant differences between the treatments.

**Table 4.** Aroma compound concentrations and odour activity values.

| Aroma Compounds * | Aroma Threshold | Aroma Series | Concentration of Aroma Compounds | | | | Odour Activity Values | | | |
|---|---|---|---|---|---|---|---|---|---|---|
| | | | GJ-P | MJ-P | GJ | MJ | GJ-P | MJ-P | GJ | MJ |
| **Acetate Esters** | | | | | | | | | | |
| Ethyl acetate (mg/L) | 7.50 | 3, 1 | 112.8 ± 8.5 a | 57.3 ± 4.0 c | 86.9 ± 5.5 b | 34.4 ± 0.2 d | 15.07 | 7.64 | 11.59 | 4.59 |
| Isobutyl acetate | 1605 | 1 | 46.9 ± 3.9 bc | 34.0 ± 3.6 c | 263.9 ± 16.3 a | 60.3 ± 0.3 b | - | - | 0.16 | - |
| 2-Methylbutyl acetate (mg/L) | 0.313 | 1 | 0.255 ± 0.012 b | 0.207 ± 0.018 b | 5.009 ± 0.145 a | 0.254 ± 0.006 b | 0.81 | 0.66 | 16.01 | 0.81 |
| Isoamyl acetate (mg/L) | 0.030 | 1 | 0.208 ± 0.009 b | 0.141 ± 0.008 b | 2.535 ± 0.169 a | 0.171 ± 0.003 b | 6.93 | 4.70 | 84.67 | 5.70 |
| Hexyl acetate | 700 | 1 | 6.73 ± 0.34 b | 5.79 ± 0.40 b | 55.24 ± 7.26 a | 4.57 ± 0.48 b | - | - | - | - |
| Octyl acetate | 50,000 | 1, 2 | 4.97 ± 0.34 b | 4.82 ± 0.15 b | 6.24 ± 0.31 a | 0.96 ± 0.05 c | - | - | - | - |
| **Volatile fatty acids** | | | | | | | | | | |
| Acetic acid (mg/L) | 200 | 3 | 139 ± 7 c | 174 ± 4 b | 172 ± 10 b | 640 ± 3 a | 0.70 | 0.87 | 0.86 | 3.20 |
| Butyric acid (mg/L) | 0.173 | 4 | 1.136 ± 0.025 a | 0.960 ± 0.017 b | 1.100 ± 0.026 a | 0.466 ± 0.003 c | 6.59 | 5.55 | 6.36 | 2.69 |
| Isobutyric acid (mg/L) | 2.30 | 4 | 2.696 ± 0.091 b | 2.363 ± 0.020 c | 3.514 ± 0.127 a | 1.571 ± 0.010 d | 1.17 | 1.03 | 1.53 | 0.68 |
| 2-Methylbutyric acid (mg/L) | 3.00 | 4 | 0.665 ± 0.087 a | 0.363 ± 0.016 b | 0.777 ± 0.027 a | 0.207 ± 0.003 c | 0.22 | 0.12 | 0.26 | - |
| Isovaleric acid (mg/L) | 0.0334 | 4 | 0.638 ± 0.032 a | 0.461 ± 0.017 c | 0.588 ± 0.006 b | 0.178 ± 0.002 d | 19.10 | 13.80 | 17.60 | 5.33 |
| Hexanoic acid (mg/L) | 0.420 | 4 | 2.45 ± 0.07 b | 2.08 ± 0.05 c | 2.63 ± 0.02 a | 0.87 ± 0.02 d | 5.83 | 4.95 | 6.26 | 2.08 |
| Octanoic acid (mg/L) | 0.500 | 4 | 1.86 ± 0.07 b | 1.49 ± 0.22 c | 3.10 ± 0.04 a | 1.25 ± 0.04 c | 3.72 | 2.98 | 6.20 | 2.50 |
| **Alcohols** | | | | | | | | | | |
| Isoamyl alcohol (mg/L) | 30.0 | 3 | 177 ± 2 a | 117 ± 5 c | 145 ± 6 b | 58 ± 1 d | 5.90 | 3.90 | 4.83 | 1.92 |
| *cis*-3-Hexen-1-ol | 1000 | 4, 5 | 50.1 ± 0.5 a | 28.3 ± 1.1 a | 41.3 ± 16.6 a | 1.0 ± 0.1 b | - | - | - | - |
| *trans*-3-Hexen-1-ol | 1000 | 5 | 22.1 ± 0.5 a | 18.9 ± 0.3 a | 20.6 ± 5.4 a | ND | - | - | - | - |
| *trans*-2-Hexen-1-ol | 1000 | 5 | 9.52 ± 3.22 a | 6.53 ± 0.34 ab | 8.08 ± 0.63 a | 2.63 ± 0.07 b | - | - | - | - |
| Hexanol (mg/L) | 1.10 | 5 | 1.180 ± 0.083 a | 1.131 ± 0.067 a | 0.657 ± 0.072 b | 0.015 ± 0.001 c | 1.07 | 1.03 | 0.60 | - |
| 1-Heptanol | 200 | 4 | 53.9 ± 6.4 a | 55.7 ± 11.0 a | 7.8 ± 0.4 b | 7.8 ± 0.4 b | 0.27 | 0.28 | - | - |
| Phenylethyl alcohol (mg/L) | 14.0 | 2 | 45.0 ± 5.0 a | 40.4 ± 0.2 ab | 38.2 ± 0.4 b | 19.0 ± 0.1 c | 3.21 | 2.89 | 2.73 | 1.36 |
| 1-Octanol | 800 | 2 | 35.8 ± 3.3 a | 31.3 ± 2.8 a | 18.0 ± 0.3 b | 14.2 ± 2.2 b | - | - | - | - |
| **Aldehydes** | | | | | | | | | | |
| Benzaldehyde | 2000 | | 12.4 ± 2.9 a | 10.7 ± 1.2 ab | 11.3 ± 1.0 ab | 6.9 ± 0.9 b | - | - | - | - |
| **Ethyl Esters** | | | | | | | | | | |
| ethyl isobutyrate | 15.0 | 1 | 27.3 ± 2.1 b | 21.7 ± 0.4 c | 41.2 ± 3.2 a | 17.7 ± 0.3 c | 1.82 | 1.45 | 2.75 | 1.18 |
| Ethyl butyrate | 400 | 1 | 422 ± 27 a | 226 ± 17 b | 406 ± 21 a | 90 ± 3 c | 1.06 | 0.57 | 1.02 | 0.23 |
| Ethyl lactate (mg/L) | 150 | 1, 4 | 8.10 ± 0.34 b | 9.23 ± 0.67 a | 9.10 ± 0.79 a | 5.99 ± 0.08 b | - | - | - | - |
| Ethyl 2-methylbutyrate | 18.0 | 1 | 3.59 ± 0.47 b | 2.43 ± 0.28 bc | 5.78 ± 0.95 a | 1.29 ± 0.05 c | 0.20 | 0.14 | 0.32 | - |
| Ethyl pentanoate | 5.00 | 1 | 1.41 ± 0.21 a | 1.38 ± 0.12 a | 1.48 ± 0.06 a | 0.95 ± 0.07 b | 0.28 | 0.28 | 0.30 | 0.19 |
| Ethyl hexanoate | 14.0 | 1, 2 | 806 ± 25 b | 496 ± 25 c | 919 ± 24 a | 228 ± 7 d | 57.57 | 35.43 | 65.64 | 16.29 |
| Ethyl heptanoate | 2.20 | 1 | 1.72 ± 0.18 b | 2.46 ± 0.27 a | 0.92 ± 0.01 c | 0.97 ± 0.08 c | 0.78 | 1.12 | 0.42 | 0.44 |
| 2-Phenylethyl acetate | 250 | 2, 5 | 12.6 ± 1.3 b | 12.1 ± 1.1 b | 244.0 ± 9.7 a | 22.7 ± 1.3 b | - | - | 0.98 | - |
| Ethyl octanoate (mg/L) | 0.580 | 1, 2 | 1.239 ± 0.064 a | 0.866 ± 0.037 b | 1.256 ± 0.006 a | 0.500 ± 0.008 c | 2.14 | 1.49 | 2.17 | 0.86 |
| Diethyl succinate | 1,200,000 | 1, 2 | 240.9 ± 28.4 a | 248.5 ± 16.9 a | 152.0 ± 13.6 b | 13.8 ± 0.3 c | - | - | - | - |
| Ethyl cinnamate | 1.10 | 5 | 0.970 ± 0.131 a | 0.747 ± 0.151 a | 0.846 ± 0.102 a | 0.141 ± 0.024 b | 0.88 | 0.68 | 0.77 | 0.13 |
| Ethyl hydrocinnamate | 1.60 | 1, 2 | 0.705 ± 0.069 a | 0.688 ± 0.114 a | 0.550 ± 0.004 a | 0.197 ± 0.052 b | 0.44 | 0.43 | 0.34 | 0.12 |
| Ethyl decanoate (mg/L) | 0.200 | 1, 3, 4 | 1.06 ± 0.04 b | 0.85 ± 0.01 c | 1.34 ± 0.08 a | 0.26 ± 0.01 d | 5.30 | 4.26 | 6.70 | 1.32 |
| **Volatile phenols** | | | | | | | | | | |
| Phenol | 5900 | 3, 5 | 3.87 ± 0.12 a | 3.01 ± 0.19 b | 2.49 ± 0.13 c | 2.57 ± 0.14 c | - | - | - | - |
| Guaiacol | 9.50 | 3, 5 | 6.26 ± 0.13 a | 5.30 ± 0.47 b | 5.27 ± 0.16 b | 4.10 ± 0.01 c | 0.66 | 0.56 | 0.55 | 0.43 |
| 4-Ethylguaiacol | 33.0 | 5 | 0.123 ± 0.028 a | 0.097 ± 0.034 ab | 0.126 ± 0.007 a | 0.043 ± 0.003 b | - | - | - | - |
| Eugenol | 6.00 | 5 | 3.02 ± 0.48 a | 1.99 ± 0.10 b | 1.81 ± 0.28 b | 0.94 ± 0.02 c | 0.50 | 0.33 | 0.30 | 0.16 |
| **C13-norisoprenoids** | | | | | | | | | | |
| β-Damascenone | 7.00 | 1, 2 | 6.57 ± 0.16 b | 2.22 ± 0.21 c | 14.11 ± 2.22 a | 0.42 ± 0.01 c | 0.94 | 0.32 | 2.01 | - |
| α-Ionone | 2.60 | 1 | 0.117 ± 0.008 b | 0.137 ± 0.009 a | 0.050 ± 0.005 c | 0.047 ± 0.000 c | - | - | - | - |
| β-Ionone | 5.00 | 1, 2 | 0.711 ± 0.021 a | 0.663 ± 0.032 a | 0.396 ± 0.016 b | 0.351 ± 0.031 b | 0.14 | 0.13 | - | - |
| **Monoterpenes** | | | | | | | | | | |
| Geraniol | 30.0 | 1, 2 | 8.59 ± 0.11 a | 4.73 ± 0.16 c | 6.10 ± 0.12 b | ND | 0.29 | 0.16 | 0.20 | - |
| Linalool | 25.2 | 2, 1 | 17.7 ± 0.5 a | 10.7 ± 0.3 c | 15.4 ± 0.6 b | 2.4 ± 0.1 d | 0.70 | 0.42 | 0.61 | - |
| Citronellol | 100 | 2 | 19.2 ± 0.8 a | 13.5 ± 0.7 b | 12.0 ± 0.5 b | 4.9 ± 0.1 c | 0.19 | 0.14 | 0.12 | - |
| Nerol | 300 | 2 | 5.16 ± 0.40 a | 3.84 ± 0.36 b | 2.92 ± 0.21 c | ND | - | - | - | - |

* Concentrations are expressed in μg/L unless otherwise noted; ND: Not Detected; different lowercase letters in rows indicate a significant difference among treatments ($p < 0.05$). GJ-P, grape juice plus pomace; MJ-P, model juice plus pomace; GJ, grape juice; MJ, model juice. Aroma Series: 1-fruity; 2-floral; 3-chemical; 4-fatty; 5-herbaceous.

When comparing GJ-P and MJ-P treatments, about half of the analysed aroma compounds did not show a significant difference between GJ-P and MJ-P treatments, which includes aldehydes, acetate esters except ethyl acetate, higher alcohols except isoamyl alco-

hol, some ethyl esters including ethyl-2-methylbutyrate, ethyl pentanoate, 2-phenylethyl acetate, diethyl succinate, ethyl cinnamate, and ethyl hydrocinnamate, 4-ethylguaiacol in volatile phenols, and β-Ionone in C13-norisoprenoids. Among these compounds, ethyl cinnamate, ethyl hydrocinnamate, and benzaldehyde were identified as important aroma compounds contributing to varietal characteristics in Pinot Noir wine [17,50]. These results indicate that grape skins largely determine the aroma profile of Pinot noir wine because MJ-P treatment showed a similar aroma profile to GJ-P treatment even in the absence of grape juice. Furthermore, most of these compounds were secondary aroma compounds, mainly originated from the metabolism of yeast, and enzyme activity during fermentation [4]. Secondary aromas are the most abundant aroma fraction in Pinot noir wine and form the aroma base of all wines [18,28].

Compared to MJ-P treatment, twenty-one compounds out of total forty-six aroma compounds analysed in this work showed significantly higher concentrations in GJ-P treatment including, monoterpenes, most of the volatile phenols, C13-norisoprenoids, volatile fatty acids except acetic acid, few ethyl esters and isoamyl alcohol. Primary aroma compounds (monoterpenes, eugenol and C13-norisoprenoids) are found in both grape skins and juice [15] hence, higher concentrations are expected in GJ-P treatment compared to MJ-P treatment. In addition, the aroma precursors originating from grape juice may also contribute to result in significantly higher concentrations of these compounds in GJ-P treatment. Previous studies have shown that precursors of monoterpenes, C13 -norisoprenoids, and volatile phenols such as eugenol are derived in the berries at earlier stages of berry development, and they have been identified as monoglucosides and diglycosides forms in both grape skin and juice [20,51]. Ethyl decanoate, ethyl octanoate, and ethyl butanoate were identified as important wine aroma compounds contributing to the fruity aroma characteristics in New Zealand Pinot noir wine [17]. Apart from the primary aromas, a higher concentration of isoamyl alcohol in GJ-P treatment may be associated with the lower YAN levels recorded in GJ-P treatment because a previous study has demonstrated that lower amounts of YAN (200–300 mg/L) led to a higher concentrations of isoamyl alcohol, 2-methyl propanol, 2-methyl butanol, and 2-phenyl ethanol in wine [52]. In addition, higher concentration of butyric acid, isobutyric acid, their corresponding ethyl esters, and branched chain fatty acids including 2-methylbutyric acid and isovaleric acid in GJ-P treatment also agrees with the lower YAN levels in wine but also has been shown that yeast strain has a profound effect on the concentration of medium chain fatty acids and their corresponding ethyl esters in wines [52,53].

When comparing GJ-P and GJ treatments, sixteen aroma compounds including acetate esters (except ethyl acetate), most of the volatile fatty acids, a few ethyl esters and β-damascenone showed significantly lower concentrations in GJ-P treatment. A previous study on Muscat blanc grapes has also shown that fermentation with skins greatly reduced ho-trienol, β-damascenone, fatty acids and esters in wine [21]. According to a previous study [5], it is possible that skins in some way inhibited the formation of these compounds by providing either competitive substrates or enzyme inhibitors or adsorbing them on their surface. It is also possible that cap management during fermentation in GJ-P and MJ-P treatments had a significant effect on aroma compound evaporation. A previous study has also shown that solid parts of the grapes inhibit the biosynthesis of volatile fatty acids in yeast cells and result in lower concentrations in the wines [54]. However, a recent study on Pinot noir grapes fermented with and without grape skins has shown fairly similar results to this study but wines showed no significant difference in most of the volatile fatty acids analysed [55]. This may be associated with the vintage differences, and higher cap management frequency used in winemaking.

Moreover, eighteen aroma compounds including all monoterpenes, C13-norisoprenoids (except β-damascenone), volatile phenols (except 4-ethylguaiacol), most of the higher alcohols, few ethyl esters, and ethyl acetate showed significantly higher concentrations in GJ-P treatment compared to GJ treatment. Most of the primary aromas, such as monoterpenes, C13-norisoprenoids, and volatile phenols (eugenol) are largely found in grape skins in free

volatile form or bound into non-volatile precursors [15,55] and hence higher concentrations are expected in GJ-P treatment. However, the higher concentration of higher alcohols in GJ-P treatment may be linked to the presence of more amino acids in the fermentation media because GJ-P treatment contained grape pomace during the fermentation. Previous studies have shown that the total production of higher alcohols increases as the concentrations of corresponding amino acids increase in grape must during fermentation [4,53,56]. Interestingly twelve aroma compounds including butyric acid, 2-methylbutyric acid, *cis*-3-hexen-1-ol, *trans*-3-hexen-1-ol, *trans*-2-hexen-1-ol, benzaldehyde, ethyl butyrate, ethyl pentanoate, ethyl octanoate, ethyl cinnamate, ethyl hydrocinnamate, and eugenol showed no significant difference between GJ-P and GJ treatments and it confirms that their concentrations are not affected by the grape solids in the media.

MJ treatment had the lowest concentration in most of the aroma compounds analysed in this study, which is mainly due to the absence of primary aroma compounds and aroma precursors which comes from grapes in MJ treatment. Hence, primary aroma compounds including C13-norisoprenoids, monoterpenes, $C_6$ alcohols (hexanol, *cis*-3-hexen-1-ol, *trans*-3-hexen-1-ol, and *trans*-2-hexen-1-ol) were not detected or detected below the limit of quantification (LOQ) in MJ treatment. MJ treatment showed a higher concentration of acetic acid (640 mg/L) compared to other treatments, which is still below the sensory threshold in red wine and well below the legal limits of acetic acid in wine [57,58]. Higher concentration of acetic acid in MJ treatment may be due to the high level of nitrogen source added in model juice, and previous studies have shown that higher YAN level in must resulted in a higher acetic acid concentration and volatile acidity in wines [52,59]. The higher acetic acid concentration in MJ treatment may also be associated with the absence of tannins and lower alcohol levels. In addition, no cap management could be another reason for relatively high acetic acid in MJ as more acetic acid is preserved in wine.

Medium-chain fatty acids ($C_6$–$C_{12}$), hexanoic acid and octanoic acid were detected in MJ treatment because they are mainly produced by yeast during sugar metabolism in fermentation [56]. Corresponding ethyl esters of medium-chain fatty acids were also found in MJ treatment such as ethyl hexanoate, ethyl heptanoate, ethyl octanoate, ethyl decanoate because medium-chain fatty acids are the initial substrate for the final formation of ethyl esters in wines [56,60]. The presence of higher alcohols in MJ treatment indicates that the amino acids in model juice have been converted to higher alcohols through the Ehrlich pathway [4]. For example, isoamyl alcohol and phenylethyl alcohol in MJ treatment is derived from leucine, and phenylalanine amino acids in the model juice, respectively. These higher alcohols are the initial substrate for the final formation of acetate esters in wine. For example, isobutyl acetate in MJ treatment is derived from the isoamyl alcohol (derived from leucine).

### 3.4.1. Odour Activity Value (OAV) Analysis

Odour activity values (OAVs) are commonly used to assess the contribution of volatile compounds to the overall wine aroma in many studies [61–64]. According to previous studies, aroma compounds with an OAV above 0.1 is considered important contributors to the overall wine aroma due to the synergistic effect of certain aroma compounds in wine [61,63]. Hence, all aroma compounds with OAV above 0.1 were considered significant aroma compounds in this study.

All the aroma series showed significant differences between treatments and the aroma series intensity patterns showed that the aroma profile of the wines mainly consisted of fruity, floral, fatty, chemical, and herbaceous aromas (Figure 1). GJ treatment showed a significantly higher total odour activity value (ΣOAV) in all aroma series except the chemical aroma series. It was mainly due to the significantly higher concentrations of 2-methylbutyl acetate, isoamyl acetate, isobutyric acid, hexanoic acid, octanoic acid, ethyl isobutyrate, and ethyl decanoate.

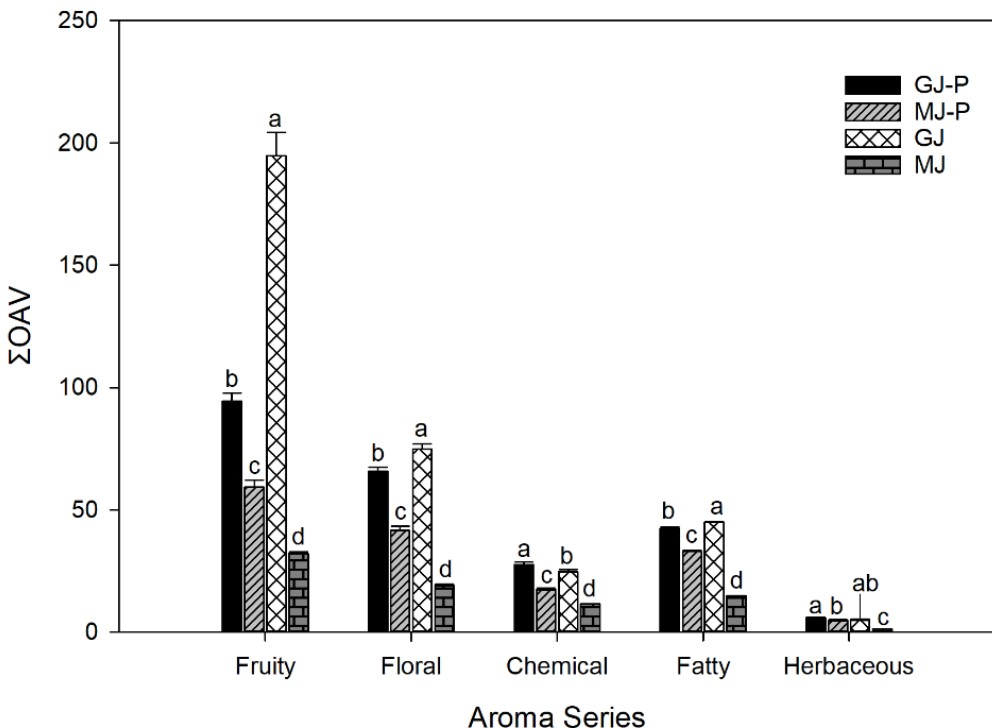

**Figure 1.** Comparison of aroma series based on ΣOAV between four treatments. Different letters in the same aroma series means a significant difference ($p < 0.05$). GJ-P, grape juice plus pomace; MJ-P, model juice plus pomace; GJ, grape juice; MJ, model juice.

The fruity aroma series was the major aroma series having the highest intensity in all the treatments, which agrees with previous observations in Pinot noir wines [28,61]. The major contributors to the fruity aroma series include two acetate esters (ethyl acetate, and isoamyl acetate) and two ethyl esters (ethyl hexanoate and ethyl decanoate) in all the wines and accounted for 85% to 90% of the ΣOAV of fruity aroma series. Previous studies also confirm that isoamyl acetate, ethyl hexanoate and ethyl decanoate are important wine aroma compounds contributing to the fruity aroma characteristics in Pinot noir wine [28,61]. When comparing GJ and GJ-P treatments, the higher ΣOAV of fruity aroma series in GJ treatment was mainly due to the higher concentrations of isoamyl acetate and 2-methylbutyl acetate. Their concentrations in the GJ treatment were twelve times and twenty times higher than that in the GJ-P treatment. This is in agreement with a previous study [55], reporting that the concentration of isoamyl acetate in the wine fermented without grape skins was eleven times higher than in the wine made with grape skins. In general, a higher concentration of acetate esters observed in GJ compared to GJ-P treatment could be due to grape skins binding or delaying the release of some volatile compounds in wine [5], and no cap management in GJ treatment during fermentation that could better preserve esters formed in wines.

Floral and fatty aroma series were also two other major aroma series in this work. For the floral aroma series, the ΣOAV was dominated by three compounds (ethyl hexanoate, phenylethyl alcohol, and ethyl octanoate) and the fatty series was dominated by six compounds (isovaleric acid, butyric acid, hexanoic acid, ethyl decanoate, octanoic acid, and isobutyric acid). However, β-damascenone was also a major contributor to the fruity aroma series in GJ treatment, while isobutyric acid was not a major contributor to the fatty aroma series in MJ treatment. Compared to the GJ-P treatment, significantly higher concentrations of ethyl hexanoate and β-damascenone in the GJ treatment led to significantly higher ΣOAV of floral aroma series in the resultant wine. A higher concentration of β-damascenone in GJ treatment indicates grape juice is a good source of β-damascenone, which can be better preserved in the wine by avoiding cap management.

### 3.4.2. Principal Component Analysis

The score plot and correlation circle plot obtained from principal component analysis (PCA) is shown in Figure 2. The ellipses in the score plot represent 95% confidence intervals for the means of the treatments. Ninety-one per cent of the differences in the concentration of aroma compounds between treatments can be explained by the first two principal components (PCs).

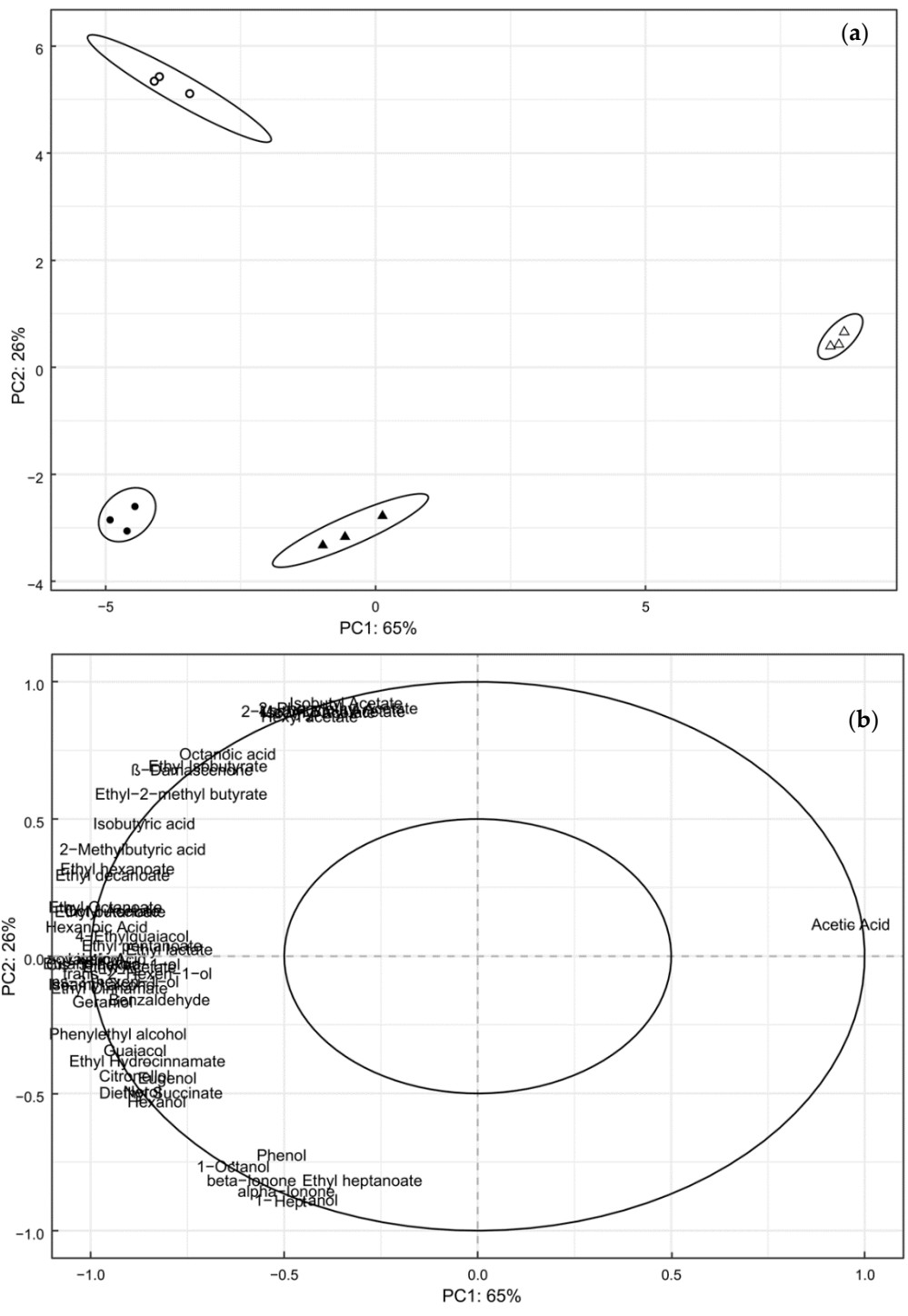

**Figure 2.** (**a**) Score-plot and (**b**) correlation circle obtained from principal component analysis of aroma compounds determined in wines, GJ-P, grape juice plus pomace (●); MJ-P, model juice plus pomace (▲); GJ, grape juice (○); MJ, model juice (△).

Triplicate ferments from each treatment were tightly clustered together on the score plot and it indicates good reproducibility of the treatments. The PC1 explained 65% of the differences and separated all four treatments. Loading vectors of PC1 and PC2 are given in Table S1. The MJ treatment was grouped away from all other treatments toward the left on the plot, showing a distinctive aroma profile. It was largely characterised by the higher concentration of acetic acid in MJ treatment. While PC2 explained 26% of the differences and wines prepared with grape pomace (GJ-P and MJ-P) were grouped separately from wines made with juice (GJ and MJ). Wines made with grape pomace were characterised mostly by primary aromas, and higher alcohols, including 1-heptanol, α-ionone, ethyl heptanoate, β-ionone, 1-octanol, phenol, hexanol, diethyl succinate, nerol, eugenol, respectively. Because it is generally accepted that most of the primary aroma compounds such as monoterpenes, C13-norisoprenoids, C6 compounds and volatile phenols (eugenol) are largely found in grape skins [15,55]. The higher concentrations of higher alcohols indicate the presence of a higher concentration of amino acids in grape pomace added treatments, which can act as aroma precursors to form higher alcohols through the Ehrlich pathway during fermentation as discussed above [4]. In contrast, GJ treatment was mainly characterised by the higher concentrations of acetate esters (except ethyl acetate) compared to grape pomace-added treatments. This may be associated with the effect of skins in GJ-P and MJ-P treatments, which may adsorb aroma compounds into skins or provide competitive substrates or enzyme inhibitors to reduce the formation of acetate esters in wines [5], and lower fermentation temperatures and aeration in GJ treatment during fermentation could enhance acetate ester concentration in wines (Figure S1) [65,66].

## 4. Conclusions

Tannin extraction is important in red wine production, and this study investigated the impact of phenolics in grape juice and polyphenols in grape pomace on tannin concentration and tannin composition. Grape skins and seeds are the most important source of tannins extracted into wine during fermentation. This study revealed the important role of phenolic compounds in grape juice in the polymerisation and composition of tannins in the wines. Inclusion of grape pomace in the fermentation enhanced the concentrations of primary aromas including monoterpenes, and C13-norisoprenoids, which are mainly responsible for floral and fruity notes in wines. The presence of corresponding aroma precursors are well documented in grape skins. Fermentation derived aroma compounds, e.g., esters, volatile fatty acids, are better preserved in the fermentation without pomace addition. These observations reflect the differences in aroma profiles between red winemaking and white winemaking.

**Supplementary Materials:** The following supporting information can be downloaded at: https://www.mdpi.com/article/10.3390/fermentation8120718/s1, Table S1: Loading vectors of first principal component (PC1) and second principal component (PC2), Figure S1: Fermentation curves: (a) total mass reduction of the ferments; (b) must temperature fluctuation during fermentation.

**Author Contributions:** Conceptualization, P.M.W., J.Z. and B.T.; methodology, P.M.W. and B.T.; software, P.M.W.; formal analysis, P.M.W.; investigation, B.T. and P.M.W.; resources, B.T.; writing—original draft preparation, P.M.W.; writing—review and editing, J.Z. and B.T.; supervision, B.T.; project administration, B.T.; funding acquisition, B.T. All authors have read and agreed to the published version of the manuscript.

**Funding:** This research received no external funding.

**Institutional Review Board Statement:** Not applicable.

**Informed Consent Statement:** Not applicable.

**Data Availability Statement:** All data can be found in the manuscript and the Supplementary Materials.

**Acknowledgments:** Authors gratefully acknowledge Jason Breitmeyer for his assistance with the GC-MS analysis, Jenny Zhao for her assistance with the HPLC analysis, and Richard Hider for his assistance with other chemical analyses.

**Conflicts of Interest:** The authors declare no conflict of interest.

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
