# Peer review of "Characterisation of Tannin and Aroma Profiles of Pinot Noir Wines Made with or without Grape Pomace"

_fermentation, doi:10.3390/fermentation8120718_

Round 1

Reviewer 1 Report

Journal: Fermentation

Manuscript ID: fermentation-2033875

Dear authors,

The manuscript fermentation-2033875 entitled “Characterisation of tannin and aroma profiles of Pinot noir wines made with and without grape pomace” is an interesting study evaluating the differences given by the presence of pomace/grape solids (i.e. skins, seeds) in phenolic compounds and aroma content and profile in the production of Pinot Noir wines. The experiment design comparing four fermentations in triplicate with/with pomace in grape juice and model juice to evaluate the influence of the different grape part (pulp, skins and seeds) is appropriate for the evaluation of the research question as well as the applied analytical methods. The results are well presented and adequately discussed.

Therefore, I suggest the publication of the manuscript with some minor modifications listed here below.

Introduction

Line 28-31: This sentence is not well connected in the speech, please to re-write it.

Line 54: “Eugenol” is the only compounds listed among classes of compounds, I suggest changing it in “volatile phenols”

Line 77: please to correct “theimpact”

Material and Method

Line 86-88: add citations or a description for the standards synthesis.

Line 95: please to report if the quantity is related to each replicate.

Line 100-106: Are the concentration referred to each independent solution, or to the final model juice? I suppose that are the final concentration of the model juice used for the fermentation, but I suggest explaining it better.

Results

Line 277-279: The interaction between phenolic compounds present in the pulp and the tannin extraction is an interesting point that must be better explain reporting some examples and previous results from scientific literature. In the authors opinion, may it be related also to the extractability of tannins from grape cell walls or to chemical reactivity after the extraction?   

Line 298: Table 2, Please to double check the reported anthocyanins. Moreover, cyanidin-3-sophoroside and delphinidin-3-rutinoside are unusual anthocyanins in Vitis vinifera grape, how did you perform the identification?

Line 319-324: Some hydroxycinnamic acids (e.g. caftaric and fertaric acids) are more present in the pulp than in the skins (https://doi.org/10.1016/j.foodres.2013.08.002), it may be some different oxidation extent in presence/absence of grape pomace?

Line 336-340: Please to enrich this explanation.

Line 472: Figure 2b is difficult to read.

Author Response

Thanks for your useful feedback and comments. We have responded to your comments point by point and revised the manuscript accordingly with changes highlighted in yellow.

Reviewer 2 Report

The manuscript discusses the contribution of grape pomace to the concentration and composition of tannins and the aroma profile of Pinot Noir wine. The studies consisted of four treatments using different fermentation media. As expected, the addition of grape pomace significantly increased the concentration of tannins in wines, but significantly reduced some important aromas such as acetate esters (except ethyl acetate), most volatile fatty acids, some ethyl esters and β-damascenone, but increased some primary aromas in wines due to the presence of their aroma precursors in the berry skin. The importance of the results is shown in the fact that grape pomace can bind or slow down the release of certain aromatic compounds and/or their loss.

A total of 51 relevant references are included in the excellently written manuscript, the results are statistically processed and graphically presented in 4 tables and 2 figures.

The only thing I miss is the unavailability of supplementary materials, both Table S1 in Figure s1!

Author Response

Thanks very much for your positive feedback on our manuscript. We have included supplementary Table and Figure in the main manuscript.